# Poor school attendance and exclusion: a systematic review protocol on educational risk factors for self-harm and suicidal behaviours

Sophie Epstein,[1,2,3] Emmert Roberts,[3,4,5] Rosemary Sedgwick,[2,3] Katie Finning,[6] Tamsin Ford,[6] Rina Dutta,[1,3,5] Johnny Downs[1,2,3]

[1]NIHR Maudsley Biomedical Research Centre, South London and Maudsley NHS Foundation Trust, London, UK
[2]Department of Child and Adolescent Psychiatry, King's College London, London, UK
[3]South London and Maudsley NHS Foundation Trust, London, UK
[4]National Addiction Centre, King's College London, London, UK
[5]Department of Psychological Medicine, King's College London, London, UK
[6]University of Exeter Medical School, Exeter, UK

**Correspondence to**
Dr Sophie Epstein;
sophie.epstein@kcl.ac.uk

## ABSTRACT

**Introduction** Schools have an important role in recognising and preventing self-harm and suicidal behaviour in their students, however little is known about which educational factors are associated with heightened risk. We will systematically review the existing evidence on two key educational performance indicators that are routinely collected by school administrative systems: school attendance and exclusion. We will investigate their association with self-harm and suicidal behaviour in school-age children and adolescents. Knowledge of this association could help inform suicide prevention strategies at clinical, school and population levels.

**Methods and analysis** We will conduct a systematic search of Medline, EMBASE, PsycINFO, British Education Index and Education Resources Information Centre (ERIC) from 1 January 1990, and conduct a manual search for additional references. We aim to identify studies that explore the association between poor school attendance or exclusion and self-harm or suicidal behaviours in school-age children and adolescents. Two independent reviewers will screen titles, abstracts and full-text documents and independently extract relevant data for analysis. Study quality will be assessed using a modified Newcastle-Ottawa Scale. A descriptive analysis will be performed, and where appropriate, results will be combined in meta-analyses.

**Ethics and dissemination** This is a systematic review of published literature, and therefore ethical approval will not be sought. We will publish reports in health and education journals, present our work at conferences focused on school mental health and communicate our findings to practitioners and managers in public health, education and child mental health.

**PROSPERO registration number** CRD42018088608.

### Strengths and limitations of this study

- ► This will be the first systematic review to investigate poor school attendance and exclusion as risk factors for self-harm and suicidal behaviour in school-age children and adolescents.
- ► The methods involve a systematic search of databases and other sources from both health and education fields, including comprehensive forward and backward citation searching and contacting key authors in the field.
- ► The wide inclusion criteria mean that this review will serve as a broad overview of the questions of interest and will allow identification of particular gaps in the existing evidence base.
- ► The use of wide inclusion criteria may also create challenges in synthesising the evidence as constructs described in each study may not be comparable.
- ► This systematic review does not include non-peer-reviewed literature and only includes studies published in the English language and therefore may not report an entirely comprehensive view of the literature on this topic.

## INTRODUCTION

Self-harm and suicidal behaviour in children and adolescents are important public health issues. International community-based studies have reported a prevalence of self-harm in adolescents of approximately 10%.[1] Suicide is the second most common cause of death worldwide for the 10–24 age group[2] and national studies in several countries have found that on average between 1% and 10% of adolescents report having attempted suicide during their lifetime.[3–7] Self-harm and attempted suicide are among the strongest predictors of completed suicide[8 9] and are strongly associated with mental disorders including depression and anxiety.[6]

In the UK, the focus of current child and adolescent mental health policy is on developing closer collaborations between the health and education sectors and supporting schools to develop capacity to play a greater role in supporting and promoting pupils' mental health.[10] This includes a role in early identification of those at risk of mental health difficulties or risky behaviours including self-harm. Currently, many school-based suicide

BMJ

prevention programmes exist, but most constitute universal prevention rather than using strategies that target particular at-risk groups.[11] In the UK, it is unclear how broadly such interventions are implemented or how at-risk groups are targeted (Evans, Parker, Russell *et al*, Adolescent self-harm prevention and intervention in secondary schools: a survey of staff in England and Wales, In Submission).

Any policy development needs to be informed by robust evidence on the factors that could help identify those children and adolescents at greater risk of self-harm or suicidal behaviours. A number of risk factors for self-harm and suicidal behaviour in children and adolescents have been identified, which include both psychological factors such as mental disorders, drug and alcohol misuse; personality characteristics such as impulsivity; as well as sociodemographic factors such as female sex for self-harm, and male sex for completed suicide, low socioeconomic status, adverse childhood experiences, family discord and bullying.[1 12 13] Risk profiles are also found to vary between subgroups when stratified by certain sociodemographic factors such as gender, age and ethnicity.[1 3 14]

A small number of studies have reported an association between suicidal phenomena and poor school attendance, actual and perceived educational attainment, poor school connectedness and experiences, and negative attitudes towards school.[1 12 15–20] There is also evidence of higher rates of suicide attempts among young adults with a history of poor school performance[21 22] but a lower risk among those who attended schools with a lower average academic performance.[22]

Beyond this however, educational factors, which may be particularly pertinent in this age group, are yet to be explored in any detail. This is surprising, given that data on these factors are routinely collected by education systems in many high-income countries.

Although many systematic reviews that explore risk factors for self-harm and suicidal behaviour among children and adolescents have been published, we have not been able to identify any which specifically report on educational risk factors, and none which attempt to comprehensively identify the literature that explores poor school attendance and school exclusion as possible risk factors for self-harm and suicidal behaviour.

In this systematic review, poor school attendance and exclusion from school are hypothesised to be two potentially important risk factors for self-harm and suicidal behaviours in school-age populations. We seek to answer the question: Is poor attendance or exclusion from school associated with self-harm and/or suicidal behaviour in school-age children and adolescents? We also aim to explore whether this association persists after accounting for other potentially confounding sociodemographic risk factors for self-harm and suicidal behaviour such as age, gender and ethnicity.

## METHODS

This protocol follows the Preferred Reporting Items for Systematic Review and Meta-Analysis Protocols (PRISMA-P)[23] guidelines (online supplementary file 1—PRISMA-P checklist) and is registered on PROSPERO, an international register of systematic reviews (ID CRD42018088608).[24] Any changes to the protocol will be recorded on PROSPERO.

### Eligibility criteria
#### Population
Eligible participants will be those aged 4–18 and attending school at the point of enrolment into the study or where poor attendance or exclusion from school and at least a proportion of the self-harm or suicidal behaviour are measured during this age range. The reason for this choice of age range is that it constitutes the school-age population in most countries.

#### Types of studies
Any observational studies (cohort, longitudinal, case–control, cross sectional) published from 1990 onwards will be included. Qualitative studies, case reports, intervention studies and comments/editorials will be excluded. Studies conducted in any country will be included, provided they are published in English. The reason for exclusion of older studies is that due to the changing nature of educational systems and social contexts within which self-harm occurs, such studies are less relevant to the present day. Only peer-reviewed papers will be included, and only those where the full text of the study is available. Grey literature will not be included.

#### Exposure and outcome variables
Any study that investigates a possible association between poor school attendance or school exclusion and self-harm or suicidal behaviour will be included. Studies will only be included if there is a comparison group. In order to be included, a study will need to report poor school attendance or exclusion as an exposure and self-harm or suicidal behaviour as an outcome. Exceptions to this rule are cross-sectional studies that consider self-harm or suicidal behaviour as an exposure and poor attendance or exclusion as an outcome, and which report results for univariate analyses.

The existing literature on self-harm and suicidal behaviour is extremely heterogeneous in terms of the terminology used. We will therefore consider a wide definition to encompass this variation, which will include any form of self-harm or thoughts about self-harm, and any form of suicidal behaviour including suicidal thoughts or ideation, plans, attempts or completed suicide. The data on this variable can be ascertained by any method including written self-reports, interviews, collateral reports or clinical or other secure records.

In terms of poor school attendance and exclusion, again a broad definition will be considered as terminology used in the international literature varies widely. Acceptable

variables will include any form of exclusion or suspension from school, temporary or permanent, over any period of time. Any authorised or unauthorised non-attendance at school (among pupils enrolled in school) including school refusal, school phobia, truancy or long-term absence due to ill health will be included. Studies using 'school dropout' as an exposure of interest will be excluded as this considered a different construct to poor school attendance. Additionally, those studies that consider attendance in relation to whether the child is or is not enrolled at school will be excluded. Attendance in relation to being on school roll or not taps a different set of issues and is of greatest relevance to populations in low/middle-income countries where some children do not attend school for financial reasons or due to lack of education provision. Any method of data collection will be acceptable, including self-report or collateral reports or secure records.

## SEARCH METHODS

The following health and education databases will be searched as follows:

► Medline (OvidSP platform).
► Embase (OvidSP platform).
► PsycINFO (OvidSP platform).
► British Education Index (EBSCO platform).
► Education Resources Information Centre (EBSCO platform).

Databases will be searched from 1 January 1990 until the present. The search strategy is described below.

Citation lists of included studies will then be checked for other relevant studies (backward citation searching), and forward citation searching will be conducted by systematically looking through lists of papers which have subsequently cited the included studies. Where possible, Web of Science will be used for this purpose, and in situations where the paper is not listed on Web of Science, Google Scholar will be used. We will also look through the reference lists of existing systematic reviews on similar topics to identify any further relevant papers. If any key journals are identified during the initial searches, we will hand-search these journals to look for additional relevant papers.

Once a final list of papers has been generated using the above methods, experts in the field will be contacted to identify any additional papers.

### Search strategy for electronic databases

The search strategies have been developed to include both keywords and thesaurus terms relevant to each respective database. For keyword searching, truncation and wildcards are used as necessary to allow for linguistic variations. Full search strategies can be found in online supplementary file 2. The included terms refer to the following concepts:

► Population (eg, child, adolescent).
► Attendance/exclusion (eg, absence, truancy, exclusion).

► Self-harm/suicidal behaviour (eg, suicidal, self-harm).

Due to the wide range of possible terms to describe self-harm, the list of terms used for this part of the search strategy has been taken from a recent review on suicide and self-harm published in the *British Journal of Psychiatry*.[25] Similarly, the terms for poor school attendance were taken from a related systematic review protocol which was developed in consultation with an information specialist as well as experts in the fields of child mental health and education.[26]

### Selection of references

Titles and abstracts generated from the searches of electronic databases will be exported to Endnote V.X8 software and duplicates removed. The references will then be screened for eligibility by two independent reviewers. Any uncertainties will be discussed between the two reviewers (SE and RS). Any discrepancies between the decisions of the two reviewers will be initially discussed between them, and if a consensus cannot be reached, a third reviewer (JD) will adjudicate. Included full texts will be obtained and screened by two reviewers and again any disagreement will be discussed with a third reviewer.

### Data extraction

Data will be extracted by two independent reviewers (SE and ER) according to an agreed data extraction form (online supplementary file 3). Data to be extracted will include: definitions of exposure and outcome variables and methods of ascertainment, covariates included in analysis, characteristics of the cohort including age and gender balance of participants, country, study design, relevant results (including effect estimates, CIs and p values, where reported) and information for the assessment of study quality/risk of bias. Where two studies report data from the same cohort, the larger sample, or if the same sample, the study with the longest follow-up will be used to ensure participants are not double counted.

### Quality assessment

Risk of bias within the included studies will be assessed using modified Newcastle-Ottawa Scale (NOS) for assessment of risk of bias in observational studies.[27] The modified scales can be found in online supplementary file 4. This will be carried out by two independent reviewers (SE and ER). As there is no original version of the NOS for cross-sectional studies, an adapted version used in previously published literature will be used.[28] Some items on the NOS have been adapted for relevance to this research question, in particular the items relating to acceptable methods of ascertainment of the exposure and outcome variables. Further key quality parameters will be assessed in addition to the NOS items as follows: appropriate sample size, appropriate statistical tests and clarity of reporting of exposure and outcome variables. Any discrepancies will be discussed between the two reviewers, and if a consensus cannot be reached, a third reviewer will adjudicate.

## Analysis

Included studies will be described according to their respective characteristics in the text and in tables, and the results of the studies will be discussed in a narrative synthesis, bringing together studies according to particular outcomes, exposures or type of participant where possible. Results will be standardised where possible by reporting or calculating ORs. If any two or more studies use sufficiently similar methodology, exposure and outcome variables, meta-analyses will be conducted. A random effects model will be used, due to the likelihood of considerable heterogeneity between studies, and we will present a summary effect with an associated 95% CI and p value. The level of statistical heterogeneity will be determined using the $I^2$ statistic. Where there is a large enough number of studies within a meta-analysis, publication bias will be assessed using funnel plots and Egger's test. An assessment of the strength of the evidence per outcome will be conducted using the Grading of Recommendations Assessment, Development and Evaluation approach adapted for non-randomised studies. Where possible, results stratified by gender will be presented and if the data within studies allow, we will also report subgroup analyses by age and ethnicity.

Findings from all studies will be reported and included in the descriptive analysis regardless of their risk of bias. The quality of included studies, as assessed by the NOS as described above, will be taken into consideration in the synthesis and analysis of the results, in terms of the strength of the evidence within each group of studies considered. We will conduct a sensitivity analysis removing any papers which score below 60% on the quality assessment scales and report a synthesis of the evidence from the remaining studies to assess whether this has an impact on the overall effect.

## Study status

Data collection was completed in August 2018 and analysis is expected to be completed during October 2018.

## Patient and public involvement

Consultation with young people, parents and teachers has been used to determine their perception of the importance of educational risk factors for self-harm and suicidal behaviours in this age group. It is based on their perspectives that this systematic review was deemed to be timely and crucial to conduct to inform further research work.

## DISCUSSION

This systematic review aims to bring together international evidence on the association between important educational risk factors and the common and important outcomes of self-harm and suicidal behaviour in children and adolescents. We are aware that all of these variables constitute broad constructs and the terminology used to describe these constructs varies widely in different contexts. It is therefore likely that there will be challenges involved in synthesising data on this topic.

An additional challenge is that studies may report poor school attendance and exclusion among many other variables considered as possible risk factors for self-harm and suicidal behaviour. It is possible therefore that terms referring to these particular exposures are not referenced in the titles or abstracts of the studies. We will minimise the risk of missing relevant studies by employing a range of search methods including consultation with experts to identify any additional papers.

Despite these challenges, this systematic review will be the first to specifically address the question of whether poor school attendance and exclusion are related to self-harm and suicidal behaviour in young people. We will rigorously assess the quality and strength of the evidence, which could generate further research questions, and could ultimately help inform suicide prevention strategies at clinical, school or population level.

In the UK and many other high-income countries, data on school attendance and exclusions are routinely collected for every child in the state education system. Additionally, compared with many other parameters collected, such as socioeconomic status and special educational needs, these are relatively objective and standardised measures, which could be used to identify those at higher risk of adverse outcomes, including self-harm and suicidal behaviour. With the opportunity for schools to play an important role in the prevention of self-harm and suicide,[10] it is crucial to know whether these routinely collected factors can be used as predictors of those at higher risk. The presence or absence of an association has important policy implications in terms of the design and implementation of targeted school-based suicide prevention interventions. If evidence on this association (or other important factors which may provide an alternative explanation for this association) is found to be lacking, an urgent need for further research will be highlighted.

## Ethics and dissemination

As this is a systematic review of published literature, ethical approval will not be sought. We will publish the protocol and our findings in peer-reviewed journals aimed at school and health commissioners, practitioners and managers. We will present our work at the growing numbers of national and international meetings focused on school mental health. This work will be used to inform our ongoing work with the Department for Education to examine educational risk factors for child and adolescent mental health outcomes, including secondary analyses of the National Pupil Database using linked health and school data.

**Contributors** SE (guarantor) co-conceived and designed the review with support from other authors, wrote the initial draft of the manuscript, approved the final version and agrees to be accountable for its content. ER, RS, KF, TF and RD contributed to the design of the review, critically revised the manuscript, approved the final version and agree to be accountable for its content. JD co-conceived and

designed the review with support from other authors, critically revised the drafts of the manuscript, approved the final version and agrees to be accountable for its content.

**Funding** SE has been employed as a National Institute of Health Research (NIHR) Academic Clinical Fellow and currently receives salary support from an MQ Data Science Award and from the Psychiatry Research Trust. ER is funded by a Medical Research Council (MRC) Addiction Research Clinical Training Fellowship. RS is employed as an NIHR Academic Clinical Fellow. KF's PhD is funded by the University of Exeter, via the MYRIAD study (Wellcome Trust 107496/Z/15/Z). RD is funded by a Clinician Scientist Fellowship (research project e-HOST-IT) from the Health Foundation in partnership with the Academy of Medical Sciences. JD received support via an MRC Clinical Research Training Fellowship (MR/L017105/1), Psychiatry Research Trust Peggy Pollak Research Fellowship in Developmental Psychiatry and from the NIHR Biomedical Research Centre at South London and Maudsley NHS Foundation Trust and King's College London. This paper represents independent research part funded by the National Institute for Health Research (NIHR) Biomedical Research Centre at South London and Maudsley NHS Foundation Trust and King's College London.

**Disclaimer** The views expressed are those of the authors and not necessarily those of the NHS, the NIHR or the Department of Health and Social Care.

**Competing interests** None declared.

**Patient consent** Not required.

**Provenance and peer review** Not commissioned; externally peer reviewed.

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
