## [Reviewer comments · BMJ Open]

ARTICLE DETAILS

TITLE (PROVISIONAL)	Poor school attendance and exclusion: a systematic review protocol on educational risk factors for self-harm and suicidal behaviours
AUTHORS	Epstein, Sophie; Roberts, Emmert; Sedgwick, Rosemary; Finning, Katie; Ford, Tamsin; Dutta, Rina; Downs, Johnny

VERSION 1 – REVIEW

REVIEWER	Shilpa Aggarwal Deakin University, Australia
REVIEW RETURNED	03-Jun-2018

GENERAL COMMENTS	It is an important review that will add to the existing literature.
---

REVIEWER	Rocio Casañas Research Department, Associació Centre Higiene Mental Les Corts, Grup CHM Salut Mental, Barcelona, Spain
REVIEW RETURNED	03-Jul-2018

GENERAL COMMENTS	Reviewer's report Title: "Poor school attendance and exclusion: a systematic review protocol on educational risk factors for self-harm and suicidal behaviours". Reviewer: Rocío Casañas Date: 3rd July 2016 Declaration of competing interests: I declare that I have no competing interests Level of interest: In relation about Level of interest, I think that this systematic review protocol is an article of importance in its field. Suicide behavior is an important public health problem for children and adolescents around the world, and all the studies that are carried out to study the risk factors to prevent it are important. Originality: I think it's an article that will provides relevant information about the knowledge of the association between "schools attendance and exclusion" with "self-harm and suicidal behavior" in school-age children and adolescents. As the authors comment, knowledge of this association could help to inform suicide prevention strategies at clinical, school and population level. Scientific rigor: This systematic review protocol follows the PRISMA checklist and is registered on PROSPERO, International prospective register of systematic reviews (it is funded by the
---

	National Institute for Health Research (NIHR)). I think it's an article developed by a rigorous methodology. I do not have any suggestion to add to the article. I believe the manuscript is ready to be published without changes.
--	--

VERSION 1 – AUTHOR RESPONSE

Dear Editor and Reviewers,

Many thanks for your valuable comments.

During the time our manuscript was under review by BMJ Open, we were able to continue the data collection for the systematic review at the pace anticipated. Therefore I have added a section in the manuscript in the methods section entitled 'Study status' and explained that data collection was completed in August, and that analysis is due to be completed next month (October 2018).

We trust that this clarifies the study's current status, but please do not hesitate to get in touch if I can answer any further questions.

I have also re-attached the supplementary materials in pdf format, added a section on patient and public involvement and changed the in-text citations for the supplementary files as requested. The supplementary files are therefore now in a different order.

Many thanks

Best wishes

Dr Sophie Epstein